# Towards Hierarchical Importance Attribution: Explaining Compositional Semantics for Neural Sequence Models

**Xisen Jin[§], Zhongyu Wei[†], Junyi Du[§], Xiangyang Xue[†], Xiang Ren[§]**

[§]University of Southern California
[†]Fudan University
{xisenjin, junyidu, xiangren}@usc.edu
{zywei, xyxue}@fudan.edu.cn

## Abstract

The impressive performance of neural networks on natural language processing tasks attributes to their ability to model complicated word and phrase compositions. To explain how the model handles semantic compositions, we study hierarchical explanation of neural network predictions. We identify non-additivity and context independent importance attributions within hierarchies as two desirable properties for highlighting word and phrase compositions. We show some prior efforts on hierarchical explanations, e.g. contextual decomposition, do not satisfy the desired properties mathematically, leading to inconsistent explanation quality in different models. In this paper, we start by proposing a formal and general way to quantify the importance of each word and phrase. Following the formulation, we propose Sampling and Contextual Decomposition (SCD) algorithm and Sampling and Occlusion (SOC) algorithm. Human and metrics evaluation on both LSTM models and BERT Transformer models on multiple datasets show that our algorithms outperform prior hierarchical explanation algorithms. Our algorithms help to visualize semantic composition captured by models, extract classification rules and improve human trust of models[1].

## 1 Introduction

Recent advances in deep neural networks have led to impressive results on a range of natural language processing (NLP) tasks, by learning latent, compositional vector representations of text data (Peters et al., 2018; Devlin et al., 2018; Liu et al., 2019b). However, interpretability of the predictions given by these complex, "black box" models has always been a limiting factor for use cases that require explanations of the features involved in modeling (*e.g.*, words and phrases) (Guidotti et al., 2018; Ribeiro et al., 2016). Prior efforts on enhancing model interpretability have focused on either constructing models with intrinsically interpretable structures (Bahdanau et al., 2015; Liu et al., 2019a), or developing post-hoc explanation algorithms which can explain model predictions without elucidating the mechanisms by which model works (Mohseni et al., 2018; Guidotti et al., 2018). Among these work, post-hoc explanation has come to the fore as they can operate over a variety of trained models while not affecting predictive performance of models. Towards post-hoc explanation, a major line of work, additive feature attribution methods (Lundberg & Lee, 2017; Ribeiro et al., 2016; Binder et al., 2016; Shrikumar et al., 2017), explain a model prediction by assigning importance scores to individual input variables. However, these methods are not ideal for explaining phrase-level importance, as phrase importance is often a non-linear combination of the importance of the words in the phrase. Contextual decomposition (CD) (Murdoch et al., 2018) and its hierarchical extension (Singh et al., 2019) go beyond the additive assumption and compute the contribution *solely made by a word/phrase* to the model prediction (*i.e.*, *individual contribution*), by decomposing the output variables of the neural network at each layer. Using the individual contribu-

---

[1]Project page: https://inklab.usc.edu/hiexpl/

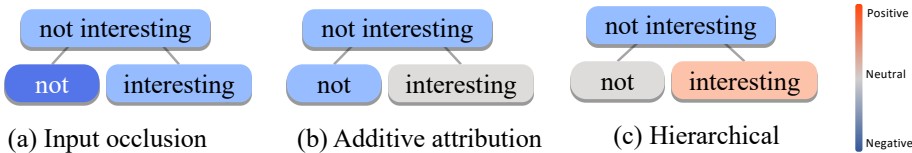

(a) Input occlusion      (b) Additive attribution      (c) Hierarchical

Figure 1: Different score attribution algorithms. (a) Input occlusion assigns a negative score for the word "interesting", as the sentiment of the phrase becomes less negative after removing "interesting" from the original sentence. (b) Additive attributions assign importance scores for words "not" and "interesting" by linearly distributing contribution score of "not interesting", exemplified with Shapley Values (Shapley, 1953). Intuitively, only (c) Hierarchical explanations highlight the negative compositional effect between the words "not" and "interesting".

tion scores so derived, these algorithms generate hierarchical explanation on how the model captures compositional semantics (*e.g.*, stress or negation) in making predictions (see Figure 1 for example).

However, despite contextual decomposition methods have achieved good results in practice, what reveals extra importance that emerge from combining two phrases has not been well studied. As a result, prior lines of work on contextual decomposition have focused on exploring model-specific decompositions based on their performance on visualizations. We identify the extra importance from combining two phrases can be quantified by studying how the importance of the combined phrase differs from the sum of the importance of the two component phrases on its own. Similar strategies have been studied in game theory for quantifying the surplus from combining two groups of players (Fujimoto et al., 2006). Following the definition above, the key challenge is to formulate the importance of a phrase on it own, *i.e.*, context independent importance of a phrase. However, we show contextual decomposition do not satisfy this context independence property mathematically.

To this end, we propose a formal way to quantify the importance of each individual word/phrase, and develop effective algorithms for generating hierarchical explanations based on the new formulation. To mathematically formalize and efficiently approximate context independent importance, we formulate $N$-*context independent importance of a phrase*, defined as the difference of model output after masking out the phrase, marginalized over all possible $N$ words surrounding the phrase in the sentence. We propose two explanation algorithms according to our formulation, namely the Sampling and Contextual Decomposition algorithm (SCD), which overcomes the weakness of contextual decomposition algorithms, and the Sampling and OCclusion algorithm (SOC), which is simple, model-agnostic, and performs competitively against prior lines of algorithms. We experiment with both LSTM and fine-tuned Transformer models to evaluate the proposed methods. Quantitative studies involving automatic metrics and human evaluation on sentiment analysis and relation extraction tasks show that our algorithms consistently outperform competitors in the quality of explanations. Our algorithms manage to provide hierarchical visualization of compositional semantics captured by models, extract classification rules from models, and help users to trust neural networks predictions.

In summary, our work makes the following contributions: (1) we identify the key challenges in generating post-hoc hierarchical explanations and propose a mathematically sound way to quantify context independent importance of words and phrases for generating hierarchical explanations; (2) we extend previous post-hoc explanation algorithm based on the new formulation of $N$-context independent importance and develop two effective hierarchical explanation algorithms; and (3) experiments demonstrate that the proposed explanation algorithms consistently outperform the compared methods (with both LSTM and Transformer as base models) over several datasets and models.

## 2   PRELIMINARIES

### 2.1   POST-HOC EXPLANATIONS OF NEURAL SEQUENCE MODELS

We consider a sequence of low-dimensional word embeddings $\mathbf{x}_{1:T} := (\mathbf{x}_1, \mathbf{x}_2, ..., \mathbf{x}_T)$, or denoted as $\mathbf{x}$ for brevity, as the input to a neural sequence model, such as standard RNNs, LSTM (Hochreiter

& Schmidhuber, 1997) and Transformers (Vaswani et al., 2017). These neural models extract hidden representations $\mathbf{h}_{1:T}$ from the input sequence $\mathbf{x}$, and feed these hidden representations to a prediction layer to generate prediction scores in the label space (*e.g.*, sentiment polarity of a sentence). For LSTM, we use the last hidden state $\mathbf{h}_T$ to give unnormalized prediction scores $s(\mathbf{x}) \in \mathbb{R}^{d_c}$ over $d_c$ label classes as follows.

$$s(\mathbf{x}) = \boldsymbol{W}_l \mathbf{h}_T, \tag{1}$$

where $\boldsymbol{W}_l \in \mathbb{R}^{d_c \times d_h}$ is a trainable weight matrix. For Transformers, the representation corresponding to the [CLS] token at the final layer is fed to the prediction layer to generate scores $s(\mathbf{x})$.

Towards post-hoc explanation of $s(\mathbf{x})$, a notable line of work, **additive feature attribution methods** (Ribeiro et al., 2016; Shrikumar et al., 2017; Sundararajan et al., 2017; Lundberg & Lee, 2017), measure word-level importance to the model prediction $s(\mathbf{x})$ by attributing an importance score $\phi(\mathbf{x}_i, \mathbf{x})$ to each word in the input sequence $\mathbf{x}_i \in \mathbf{x}$. However, the additive assumption hinders these methods from explaining the complex interactions between words and compositional semantics in a sentence (*e.g.*, modeling negation, transition, and emphasis in sentiment classification), as shown in Figure 1.

## 2.2 HIERARCHICAL EXPLANATIONS VIA CONTEXTUAL DECOMPOSITION

To capture non-linear compositional semantics, the line of work on contextual decomposition (CD) (Murdoch et al., 2018) designs non-additive measures of importance from individual words/phrases to the model predictions, and further extend to agglomerative contextual decomposition (ACD) algorithm (Singh et al., 2019) for generating hierarchical explanations.

Given a phrase $\mathbf{p} = \mathbf{x}_{i:j}$ in the input sequence $\mathbf{x}$, contextual decomposition (CD) attributes a score $\phi(\mathbf{p}, \mathbf{x})$ as the contribution *solely* from $\mathbf{p}$ to the model's prediction $s(\mathbf{x})$. Note that $\phi(\mathbf{p}, \mathbf{x})$ does not equal to the sum of the scores of each word in the phrase, *i.e.*, $\phi(\mathbf{p}, \mathbf{x}) \neq \sum_{\mathbf{x}_i \in \mathbf{p}} \phi(\mathbf{x}_i, \mathbf{x})$. Starting from the input layer, CD iteratively decomposes each hidden state $\mathbf{h}$ of the model into the contribution solely made by $\mathbf{p}$, denoted as $\boldsymbol{\beta}$, and the contributions involving the words outside the phrase $\mathbf{p}$, denoted as $\boldsymbol{\gamma}$, with the relation $\mathbf{h} = \boldsymbol{\beta} + \boldsymbol{\gamma}$ holds. The algorithm also keeps the contribution from the bias term, denoted as $\boldsymbol{\zeta}$, temporally before element-wise multiplication.

For a linear layer $\mathbf{h} = \boldsymbol{W}_i \mathbf{x}_t + \boldsymbol{b}_i$ with input $\mathbf{x}_t$, when $\mathbf{x}_t$ lies in the given phrase $\mathbf{p}$, the contribution solely from $\mathbf{x}_t$ to $\mathbf{h}$ is defined as $\boldsymbol{\beta} = \boldsymbol{W}_i \mathbf{x}_t$ when $\mathbf{x}_t$ is part of the phrase (*i.e.*, $\mathbf{x}_t \in \mathbf{p}$), and the contribution involving other words in the sentences (denoted as $\mathbf{x} \backslash \mathbf{p}$) is defined as $\boldsymbol{\gamma} = 0$. The contribution of the bias term $\boldsymbol{\zeta}$ is thus $\mathbf{b}_i$. When $\mathbf{x}_t$ lies outside of the phrase, $\boldsymbol{\gamma}$ is quantified as $\boldsymbol{W}_i \mathbf{x}_t$ and $\boldsymbol{\beta}$ is 0. In the cases when CD encounters element-wise multiplication operations $\mathbf{h} = \mathbf{h}_a \odot \mathbf{h}_b$ (*e.g.*, in LSTMs), it eliminates the multiplicative interaction terms which involve the information outside of the phrase $\mathbf{p}$. Specifically, suppose that $\mathbf{h}_a$ and $\mathbf{h}_b$ have been decomposed as $\mathbf{h}_a = \boldsymbol{\beta}^a + \boldsymbol{\gamma}^a + \boldsymbol{\zeta}^a$ and $\mathbf{h}_b = \boldsymbol{\beta}^b + \boldsymbol{\gamma}^b + \boldsymbol{\zeta}^b$, CD computes the $\boldsymbol{\beta}$ term for $\mathbf{h}_a \odot \mathbf{h}_b$ as $\boldsymbol{\beta} = \boldsymbol{\beta}^a \odot \boldsymbol{\beta}^b + \boldsymbol{\beta}^a \odot \boldsymbol{\zeta}^b + \boldsymbol{\zeta}^a \odot \boldsymbol{\beta}^b$.

When dealing with non-linear activation $\mathbf{h}' = \sigma(\mathbf{h})$, CD computes the contribution solely from the phrase $\mathbf{p}$ as the average activation differences caused $\boldsymbol{\beta}$ supposing $\boldsymbol{\gamma}$ is present or absent,

$$\boldsymbol{\beta}' = \frac{1}{2}[\sigma(\boldsymbol{\beta} + \boldsymbol{\gamma} + \boldsymbol{\zeta}) - \sigma(\boldsymbol{\gamma} + \boldsymbol{\zeta})] + \frac{1}{2}[\sigma(\boldsymbol{\beta} + \boldsymbol{\zeta}) - \sigma(\boldsymbol{\zeta})]. \tag{2}$$

Following the strategies introduced above, CD decomposes all the intermediate outputs starting from the input layer, until reaching the final output of the model $\mathbf{h}_T = \boldsymbol{\beta} + \boldsymbol{\gamma}$. The logit score $\boldsymbol{W}_l \boldsymbol{\beta}$ is treated as the contribution of the given phrase $\mathbf{p}$ to the final prediction $s(\mathbf{x})$.

As a follow-up study, Singh et al. (2019) extends CD algorithm to other families of neural network architectures, and proposes agglomerative contextual decomposition algorithm (ACD). The decomposition of activation functions is modified as $\boldsymbol{\beta}' = \sigma(\boldsymbol{\beta})$. For a linear layer $\mathbf{h}' = \boldsymbol{W}\mathbf{h} + \mathbf{b}$ with its decomposition $\mathbf{h} = \boldsymbol{\beta} + \boldsymbol{\gamma}$, the bias term $\boldsymbol{b}$ is decomposed proportionally and merged into the $\boldsymbol{\beta}'$ term of $\mathbf{h}'$, based on $\boldsymbol{\beta}' = \boldsymbol{W}\boldsymbol{\beta} + |\boldsymbol{W}\boldsymbol{\beta}|/(|\boldsymbol{W}\boldsymbol{\beta}| + |\boldsymbol{W}\boldsymbol{\gamma}|) \cdot \boldsymbol{b}$.

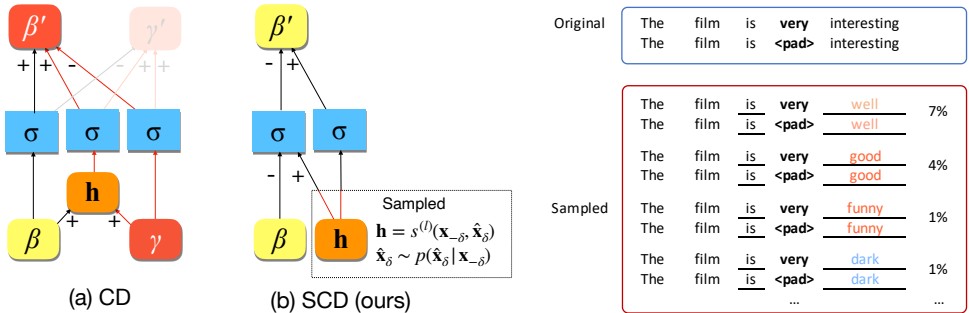

Figure 2: (Left) Illustration of the CD and SCD at calculating the decomposition for $\mathbf{h} = \sigma(\boldsymbol{\beta} + \boldsymbol{\gamma})$, following Eq. 2. Red lines indicate computation that make CD explanations dependent on the words outside the give phrase. (Right) Illustration of the sampling step $\hat{\mathbf{x}}_\delta \sim p(\hat{\mathbf{x}}_\delta | \mathbf{x}_{-\delta})$ for calculating the importance of the word *very* in SOC and SCD, with size of context $N = 1$. The padding operation is for SOC.

## 3 METHODOLOGY

In this section, we start by identifying desired properties of phrase-level importance score attribution for hierarchical explanations. We propose a measure of context-independent importance of a phrase and introduce two explanation algorithms instantiated from the proposed formulation.

### 3.1 PROPERTIES OF IMPORTANCE ATTRIBUTION FOR HIERARCHICAL EXPLANATION

Despite the empirical success of CD and ACD, no prior works analyze what common properties a score attribution mechanism should satisfy to generate hierarchical explanations that reveal compositional semantics formed between phrases. Here we identify two properties that an attribution method should satisfy to generate informative hierarchical explanations.

**Non-additivity.** Importance of a phrase $\phi(\mathbf{p}, \mathbf{x})$ should not be a sum over the importance scores of all the component words $\mathbf{x}_i \in \mathbf{p}$, *i.e.*, $\phi(\mathbf{p}, \mathbf{x}) \neq \sum_{\mathbf{x}_i \in \mathbf{p}} \phi(\mathbf{x}_i, \mathbf{x})$, in contrast to the family of additive feature attribution methods. The property is also suggested by Murdoch et al. (2018).

**Context Independence.** For deep neural networks, when two phrases combine, their importance to predicting a class may greatly change. The surplus by combining two phrases can be quantified by the difference between the importance of the combined phrase and the sum of the importance of two phrases on its own. It follows the definition of marginal interactions (Fujimoto et al., 2006) in game theory. According to the definition, the importance of two component phrases should be *at least* evaluated independently to each other. Formally, if we are only interested in how combining two phrases $\mathbf{p}_1$ and $\mathbf{p}_2$ contribute to a specific prediction for an input $\mathbf{x}$, we expect for input sentences $\tilde{\mathbf{x}}$ where only $\mathbf{p}_2$ is replaced to another phrase, the importance attribution for $\mathbf{p}_1$ remains the same, i.e., $\phi(\mathbf{p}_1, \mathbf{x}) = \phi(\mathbf{p}_1, \tilde{\mathbf{x}})$. In our bottom-up hierarchical explanation setting, we are interested in how combining a phrase and any other contextual words or phrases in the input $\mathbf{x}$ contribute. Therefore, we expect $\phi(\mathbf{p}, \mathbf{x}) = \phi(\mathbf{p}, \tilde{\mathbf{x}})$ given the phrase $\mathbf{p}$ in two different contexts $\mathbf{x}$ and $\tilde{\mathbf{x}}$.

**Limitations of CD and ACD.** Unfortunately, while CD and ACD try to construct decomposition so that $\boldsymbol{\beta}$ terms represent the contributions solely from a given a phrase, the assigned importance scores by these algorithms do not satisfy the context independence property mathematically. For CD, we see the computation of $\boldsymbol{\beta}$ involves the $\boldsymbol{\gamma}$ term of a specific input sentence in Eq. 2 (see Figure 2(a) for visualization). Similarly, for ACD, the decomposition of the bias term involves the $\boldsymbol{\gamma}$ terms of a specific input sentence. As a result, the $\boldsymbol{\beta}$ terms computed by both algorithms depend on the context of the phrase $\mathbf{p}$.

Given the limitation of prior works, we start by formulating a importance measure of phrases that satisfies both non-additivity and context independence property.

### 3.2 CONTEXT-INDEPENDENT IMPORTANCE

Given a phrase $\mathbf{p} := \mathbf{x}_{i:j}$ appearing in a specific input $\mathbf{x}_{1:T}$, we first relax our setting and define the importance of a phrase independent of contexts of length $N$ adjacent to it. The *N-context independent importance* is defined as the output difference after masking out the phrase $\mathbf{p}$, marginalized over all the possible contexts of length $N$, denoted as $\hat{\mathbf{x}}_{\delta}$, around $\mathbf{p}$ in the input $\mathbf{x}$. For an intuitive example, to evaluate the context independent importance up to one word of *very* in the sentence *The film is very interesting* in a sentiment analysis model, we sample some possible adjacent words before and after the word *very*, and average the prediction difference after some practice of masking the word *very*. Figure 2 illustrates an example for the sampling and masking steps. The $N$-context independent importance is formally written as,

$$\phi(\mathbf{p}, \hat{\mathbf{x}}) = \mathbb{E}_{\mathbf{x}_{\delta}}[s(\mathbf{x}_{-\delta}; \hat{\mathbf{x}}_{\delta}) - s(\mathbf{x}_{-\delta} \backslash \mathbf{p}; \hat{\mathbf{x}}_{\delta})], \tag{3}$$

where $\mathbf{x}_{-\delta}$ denotes the resulting sequence after masking out a context of length $N$ surrounding the phrase $\mathbf{p}$ from the input $\mathbf{x}$. Here, $\hat{\mathbf{x}}_{\delta}$ is a sequence of length $N$ sampled from a distribution $p(\hat{\mathbf{x}}_{\delta}|\mathbf{x}_{-\delta})$, which is conditioned on the phrase $\mathbf{p}$ as well as other words in the sentence $\mathbf{x}$. Accordingly, we use $s(\mathbf{x}_{-\delta}; \hat{\mathbf{x}}_{\delta})$ to denote the model prediction score after replacing the masked-out context $\mathbf{x}_{-\delta}$ with a sampled context $\hat{\mathbf{x}}_{\delta}$. We use $\mathbf{x} \backslash \mathbf{p}$ to denote the operation of masking out the phrase $\mathbf{p}$ from the input sentence $\mathbf{x}$. The specific implementation of this masking out operation varies across different explanation algorithms.

Following the notion of $N$-context independent importance, we define *context-independent importance* of a phrase $\mathbf{p}$ by increasing the size of the context $N$ to sufficiently large (*e.g.*, length of the sentence). The context independent importance can be equivalently written as follows.

$$\phi^g(\mathbf{p}) = \mathbb{E}_{\mathbf{x}}[s(\mathbf{x}) - s(\mathbf{x} \backslash \mathbf{p})|\mathbf{p} \subseteq \mathbf{x}]. \tag{4}$$

While it is possible to compute the expectations in Eqs. 3 and 4 by sampling from the training set, it is common that a phrase occurs sparsely in the corpus. Therefore, we approximate the $N$-context independent importance by sampling from a language model pretrained on the training corpus. The language model helps model a smoothed distribution of $p(\hat{\mathbf{x}}_{\delta}|\mathbf{x}_{-\delta})$. In practice, all our explanation algorithms make use of $N$-context independent importance following Eq. 3, where the size of the neighborhood $N$ is a parameter to be specified to approximate the context independent importance.

### 3.3 SAMPLING AND CONTEXTUAL DECOMPOSITION ALGORITHM

In CD, the desirable *context independence property* is compromised when computing decomposition of activation functions, as discussed in Section 3.1. Following the new formulation of context-independent importance introduced in Section 3.2, we present a simple modification of the contextual decomposition algorithm, and develop a new *sampling and contextual decomposition* (SCD) algorithm for effective generation of hierarchical explanations.

SCD is a layer-wise application of our formulation. The algorithm only modifies the way to decompose activation functions in CD. Specifically, given the output $\mathbf{h} = s^{(l)}(\mathbf{x})$ at an intermediate layer $l$ with the decomposition $\mathbf{h} = \boldsymbol{\beta} + \boldsymbol{\gamma}$, we decompose the activation value $\sigma(\mathbf{h})$ into $\boldsymbol{\beta}' + \boldsymbol{\gamma}'$, with the following definition:

$$\boldsymbol{\beta}' = \mathbb{E}_{\boldsymbol{\gamma} \sim p(\boldsymbol{\gamma}|\mathbf{x}_{-\delta})}[\sigma(\boldsymbol{\beta} + \boldsymbol{\gamma}) - \sigma(\boldsymbol{\gamma})] = \mathbb{E}_{\mathbf{h} \sim p(\mathbf{h}|\mathbf{x}_{-\delta})}[\sigma(\mathbf{h}) - \sigma(\mathbf{h} - \boldsymbol{\beta})], \tag{5}$$

*i.e.*, $\boldsymbol{\beta}'$ is defined as the expected difference between the activation values when the $\boldsymbol{\beta}$ term is present or absent. $\mathbf{h}$ is computed for different input sequences $\mathbf{x}$ with the contexts of the phrase $\mathbf{p}$ sampled from the distribution $p(\hat{\mathbf{x}}_{\delta}|\mathbf{x}_{-\delta})$. Eq. 5 is a layer wise application of Eq. 4, where the masking operation is implemented with calculating $\sigma(\mathbf{h} - \boldsymbol{\beta})$. Figure 2(b) provides a visualization for the decomposition.

**Algorithmic Details.** To perform sampling, we first pretrain a LSTM language model from two directions on the training data. During sampling, we mask the words that are not conditioned in $p(\hat{\mathbf{x}}_{\delta}|\mathbf{x}_{-\delta})$. Some other sampling options include performing Gibbs sampling from a masked language model (Wang et al., 2019). The algorithm then obtain a set of samples $\mathcal{S}$ by sampled from the language model. For each sample in $\mathcal{S}$, the algorithm records the input of the $i$-th non-linear activation function to obtain a sample set $\mathcal{S}_{\mathbf{h}}^{(i)}$. During the explanation, the $\boldsymbol{\beta}$ term of the $i$-th non-linear

activation function is calculated as,

$$\boldsymbol{\beta}' = \frac{1}{|\mathcal{S}_{\mathbf{h}}^{(i)}|} \sum_{\mathbf{h} \in \mathcal{S}_{\mathbf{h}}^{(i)}} [\sigma(\mathbf{h}) - \sigma(\mathbf{h} - \boldsymbol{\beta})]. \tag{6}$$

Some neural models such as Transformers involve operations that normalize over different dimensions of a vectors, e.g. layer normalization operations. We observe improved performance by not decomposing the normalizer of these terms when the phrase $\mathbf{p}$ is shorter than a threshold, assuming that the impact of $\mathbf{p}$ to the normalizer can be ignored. Besides, for element-wise multiplication in LSTM models, we treat them in the same way as other non-linear operations and decompose them as Eq. 5, where the decomposition of $\mathbf{h}_1 \odot \mathbf{h}_2$ is written as $\boldsymbol{\beta}' = \mathbb{E}_{\boldsymbol{\gamma}_1, \boldsymbol{\gamma}_2}[(\boldsymbol{\beta}_1 + \boldsymbol{\gamma}_1) \odot (\boldsymbol{\beta}_2 + \boldsymbol{\gamma}_2) - \boldsymbol{\gamma}_1 \odot \boldsymbol{\gamma}_2]$. We avoid decomposing softmax functions.

### 3.4 SAMPLING AND OCCLUSION ALGORITHM

We show it is possible to fit input occlusion (Li et al., 2016) algorithm into our formulation. Input occlusion algorithm calculate the importance of $\mathbf{p}$ specific to an input sentence $\mathbf{x}$ by observing the prediction difference caused by replacing the phrase $\mathbf{p}$ with padding tokens, noted as $\mathbf{0_p}$,

$$\phi(\mathbf{p}, \mathbf{x}) = s(\mathbf{x}) - s(\mathbf{x_{-p}}; \mathbf{0_p}) \tag{7}$$

It is obvious that the importance score obtained by the input occlusion algorithm is dependent on the all the context words of $\mathbf{p}$ in $\mathbf{x}$. To eliminate the dependence, we sample the context around the phrase $\mathbf{p}$. This leads to the Sampling and Occlusion (SOC) algorithm, where the importance of phrases is defined as the expected prediction difference after masking the phrase for each replacement of contexts.

**Algorithmic Details.** Similar to SCD, SOC samples neighboring words $\hat{\mathbf{x}}_\delta$ from a trained language model $p(\hat{\mathbf{x}}_\delta | \mathbf{x}_{-\delta})$ and obtain a set of neighboring word replacement $\mathcal{S}$. For each replacement $\hat{\mathbf{x}}_\delta \in \mathcal{S}$, the algorithm computes the model prediction differences after replacing the phrase $\mathbf{p}$ with padding tokens. The importance $\phi(\mathbf{p}, \mathbf{x})$ is then calculated as the average prediction differences. Formally, the algorithm calculates,

$$\phi(\mathbf{p}, \mathbf{x}) = \frac{1}{|\mathcal{S}|} \sum_{\hat{\mathbf{x}}_\delta \in \mathcal{S}} [s(\mathbf{x}_{-\delta}; \hat{\mathbf{x}}_\delta) - s(\mathbf{x}_{-\{\delta, \mathbf{p}\}}; \hat{\mathbf{x}}_\delta; \mathbf{0_p})]. \tag{8}$$

Sampling and Occlusion is advantageous in that it is model-agnostic and easy to implement. The input occlusion algorithm inside Eq. 8 can also be replaced with other measure of phrase importance, such as Shapley values (Shapley, 1953), with the phrase $\mathbf{p}$ and other input words considered as players. We expect it is helpful for longer sequences when there are multiple evidences outside the context region saturating the prediction.

## 4 EXPERIMENTS

We evaluate explanation algorithms on both shallow LSTM models and deep fine-tuned BERT Transformer (Devlin et al., 2018) models. We use two sentiment analysis datasets, namely the Stanford Sentiment Treebank-2 (SST-2) dataset (Socher et al., 2013) and the Yelp Sentiment Polarity dataset (Zhang et al., 2015), as well as TACRED relation extraction dataset (Zhang et al., 2017) for experiments. The two tasks are modeled as binary and multi-class classification tasks respectively. For the SST-2 dataset, while it provides sentiment polarity scores for all the phrases on the nodes of the constituency parsing trees, we do *not* train our model on these phrases, and use these scores as the evaluation for the phrase level explanations. Our Transformer model is fine-tuned from pretrained BERT (Devlin et al., 2018) model. See Appendix A for other implementation details.

**Compared Methods.** We compare our explanation algorithm with following baselines: Input occlusion (Li et al., 2016) and Integrated Gradient+SHAP (GradSHAP) (Lundberg & Lee, 2017); two algorithms applied for hierarchical explanations, namely Contextual Decomposition (CD) (Murdoch et al., 2018), and Agglomerative Contextual Decomposition (ACD) (Singh et al., 2019). We

| Dataset | SST-2 | | | | Yelp Polarity | | TACRED | |
|---|---|---|---|---|---|---|---|---|
| Model | BERT | | LSTM | | BERT | LSTM | BERT | LSTM |
| Metric | word $\rho$ | phrase $\rho$ | word $\rho$ | phrase $\rho$ | word $\rho$ | word $\rho$ | word $\rho$ | word $\rho$ |
| Input Occlusion | 0.2229 | 0.4081 | 0.6489 | 0.4899 | 0.3781 | 0.6935 | 0.7646 | 0.5756 |
| Direct Feed | 0.2005 | 0.4889 | 0.6798 | 0.5588 | 0.3875 | 0.7905 | 0.1986 | 0.5771 |
| GradSHAP | 0.5073 | 0.5991 | 0.7024 | 0.5402 | 0.5791 | 0.7388 | 0.2965 | 0.6651 |
| CD | 0.2334 | 0.3068 | 0.6231 | 0.4727 | 0.2645 | 0.7451 | 0.0052 | 0.6508 |
| ACD | 0.3053 | 0.3698 | 0.2495 | 0.1856 | 0.3010 | 0.5024 | 0.2027 | 0.0291 |
| Statistic | 0.5223 | 0.4741 | **0.7271** | 0.4959 | **0.7294** | **0.9094** | 0.5324 | **0.7662** |
| SCD | 0.5481 | 0.6015 | 0.7151 | **0.5664** | 0.7180 | 0.7793 | 0.7980 | 0.6823 |
| SOC | **0.6265** | **0.6628** | 0.7226 | 0.5649 | 0.6971 | 0.7683 | **0.7982** | 0.7354 |

Table 1: Correlation between word & phrase importance attribution and linear model coefficients & SST-2 human annotations, achieved by baselines and our explanation algorithms.

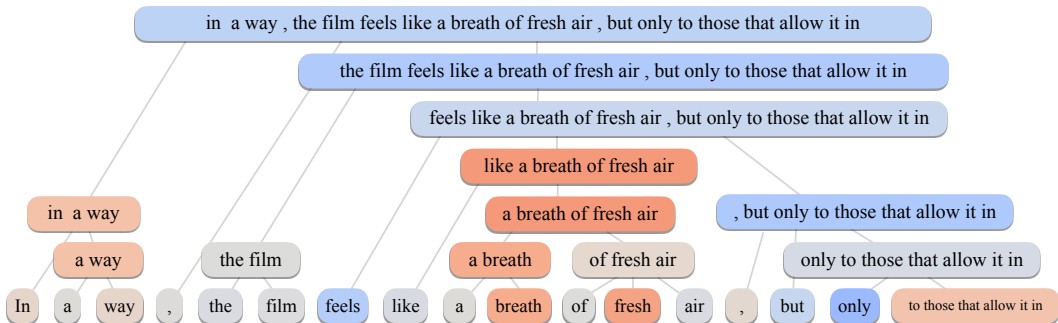

Figure 3: Hierarchical Explanation of a prediction by the BERT Transformer model on SST-2. We generate explanations for all the phrases on the truncated constituency parsing tree, with positive sentiments shown in red and negative sentiments shown in blue. We see our method identify positive segments in the overall negative sentence, such as "a breath of fresh air"

also compare with a naive however neglected baseline in prior literature, which directly feed the given phrase to the model and take the prediction score as the importance of the phrase, noted as Direct Feed. In BERT models, Direct Feed is implemented by turning off the attention mask of words except the `[CLS]` token and the phrase to be explained. For our algorithms, we list the performance of corpus statistic based approach (Statistic) for approximating context independent importance in Eq. 3, Sampling and Contextual Decomposition (SCD), and Sampling and Occlusion (SOC) algorithm. In section 4.4, we also consider padding instead of sampling the context words in SCD and SOC.

## 4.1 HIERARCHICAL VISUALIZATION OF IMPORTANT WORDS AND PHRASES

We verify the performance of our algorithms in identifying important words and phrases captured by models. We follow the quantitative evaluation protocol proposed in CD algorithm (Murdoch et al., 2018) for evaluating word-level explanations, which computes Pearson correlation between the coefficients learned by a linear bag-of-words model and the importance scores attributed by explanation methods, also noted as the **word** $\rho$. When the linear model is accurate, its coefficients could stand for general importance of words. For evaluating phrase level explanations, we notice the SST-2 dataset provides human annotated real-valued sentiment polarity for each phrase on constituency parsing trees. We generate explanations for each phrase on the parsing tree and evaluate the Pearson correlation between the ground truth scores and the importance scores assigned for phrases, also noted as the **phrase** $\rho$. We draw $K = 20$ samples for $N = 10$ words adjacent to the phrase to be explained at the sampling step in our SOC and SCD algorithms. The parameter setting is trade-off between the efficiency and performance. See Section 4.4 for detailed parameter analysis.

Table 1 shows word $\rho$ and phrase $\rho$ achieved by our algorithms and competitors. Generally, explanation algorithms that follow our formulations achieve highest word $\rho$ and phrase $\rho$ for all the datasets and models. SOC and SCD perform robustly on the deep Transformer model, achieving higher word

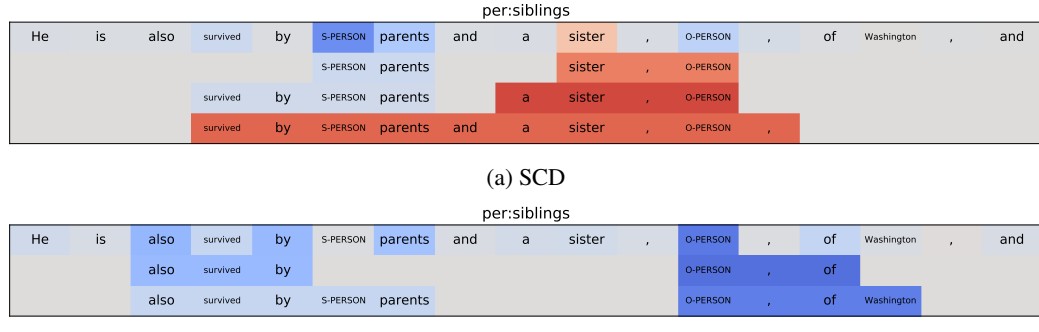

(a) SCD

(b) CD

Figure 4: Extracting phrase-level classification patterns from LSTM relation extraction model with SCD. Red indicate evidence for predicting the class, and blue indicate distractor for predicting the class. By applying the agglomerative clustering algorithm and defining a threshold score, we effecively extract "a sister, O-Person" as a classification rule for the relation per:siblings. However, we see CD fails in this example.

$\rho$ and phrase $\rho$ than input occlusion and contextual decomposition algorithms by a large margin. We see the simple Direct Feed method provide promising results on shallow LSTM networks, but fail in deeper Transformer models. The statistic based approximation of the context independent importance, which do not employ a trained sampler, yields competitive words $\rho$ but low phrase $\rho$. Our analysis show that it is common that a long phrase does not exist in previously seen examples. In this case, phrase $\rho$ achieved by statistic based approximation is pushed towards that of the input occlusion algorithm.

Qualitative study also shows that our explanation visualize complicated compositional semantics captured by models, such as positive segments in the negative example, and adversative conjunctions connected with "but". We present an example explanation provided by SOC algorithm in Figure 3 and Appendix.

## 4.2 Explanation as Classification Pattern Extraction from Models

We show our explanation algorithm is a nature fit for extracting phrase level classification rules from neural classifiers. With the agglomerative clustering algorithm in Singh et al. (2019), our explanation effectively identify phrase-level classification patterns without evaluating all possible phrases in the sentence even when a predefined hierarchy does not exist. Figure 4 show an example of automatically constructed hierarchy and extracted classification rules in an example in the TACRED dataset.

## 4.3 Enhancing Human Trust of Models

We follow the human evaluation protocol in Singh et al. (2019) and study whether our explanations help subjects to better trust model predictions. We ask subjects to rank the provided visualizations based on how they would like to trust the model. For the SST-2 dataset, we show subjects the predictions of the fine-tuned BERT model, and the explanations generated by SOC, SCD, ACD and GradSHAP algorithms for phrases. The phrase polarities are visualized in a hierarchy with the provided parsing tree of each sentence in the dataset. For the TACRED dataset, we show the explanations provided by SOC, SCD, CD and Direct Feed algorithms on the LSTM model. We binarilize the importance of a phrase by calculating the difference between its importance to the predicted class and the its top importance to other classes, and the hierarchies are constructed automatically with agglomerative clustering (Singh et al., 2019). Figure 5 shows average ranking score of explanations, where 4 for the best, and 1 for the worst. On the SST-2 dataset, SOC achieves significantly higher ranking score than ACD and GradSHAP, showing a $p$-value less than 0.05 and 0.001 respectively. On the TACRED dataset, SCD achieve the best ranking score, showing significantly better ranking score than CD and Direct Feed with a $p$-value less than $10^{-6}$ .

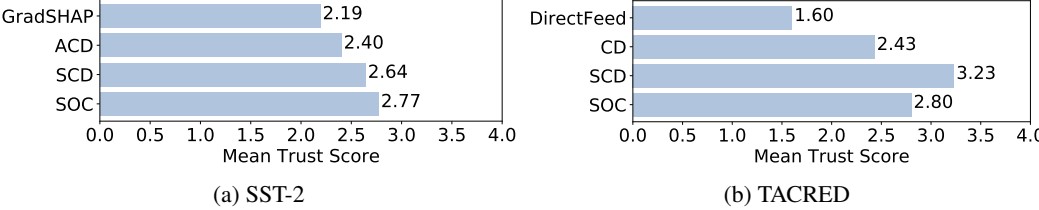

| (a) SST-2 | (b) TACRED |
|---|---|

Figure 5: Results for human evaluation on the Transformer model trained on SST-2 sentiment analysis dataset (between SOC, SCD, ACD, GradSHAP) and the LSTM model trained on TACRED relation extraction dataset (between SOC, SCD, CD, DirectFeed).

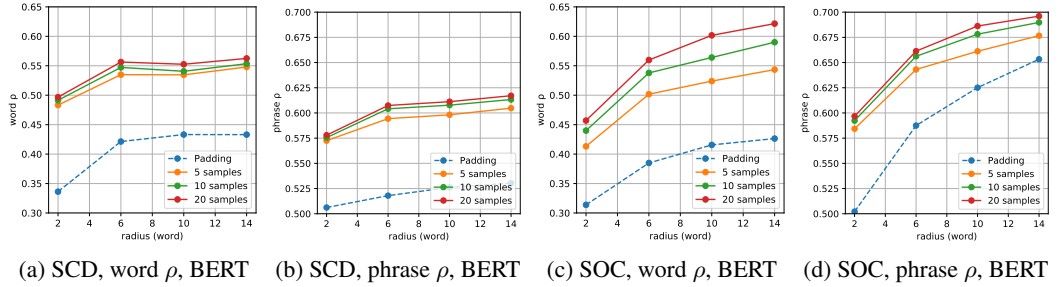

(a) SCD, word $\rho$, BERT    (b) SCD, phrase $\rho$, BERT    (c) SOC, word $\rho$, BERT    (d) SOC, phrase $\rho$, BERT

Figure 6: Word $\rho$ and phrase $\rho$ curves as the size of the context region $N$ and the number of samples $K$ change on the BERT model trained on the SST-2 dataset. Dash line notes for the performance of padding the context words instead of sampling.

## 4.4 PARAMETER ANALYSIS

Both SOC and SCD algorithms require specifying the size of the context region $N$ and the number of samples $K$. In Figure 6 (also Figure 7 in Appendix) we show the impact of these parameters. We also plot the performance curves when we pad the contexts instead of sampling. We see sampling the context achieves much better performance than padding the context given a fixed size of the context region $N$. We also see word $\rho$ and phrase $\rho$ increase as the number of samples $K$ increases. The overall performance also increases as the size of the context region $N$ increases at the early stage, and saturates when $N$ grows large, which is consistent with our hypothesis that words or phrases usually do not interact with the words that are far away them in the input.

## 5 RELATED WORKS

Interpretability of neural networks has been studied with vairous techniques, including probing learned features with auxiliary tasks (Tenney et al., 2019), or designing models with inherent interpretability (Bahdanau et al., 2015; Lei et al., 2016). A major line of work, local explanation algorithms, explains predictions by assigning importance scores for input features. This line of work include input occlusion (Kádár et al., 2017), gradient based algorithms (Simonyan et al., 2013; Hechtlinger, 2016; Ancona et al., 2017), additive feature attribution methods (Ribeiro et al., 2016; Shrikumar et al., 2017; Sundararajan et al., 2017), among which Shapley value based approaches (Lundberg & Lee, 2017) have been studied intensively because of its good mathematical properties. In the context of input occlusion explanations, researchers also study how to efficiently marginalize over input features to be explained (Zintgraf et al., 2017; Chang et al., 2019), while our research show extra focus could be placed on marginalizing over contexts. Regarding explanations of models with structured inputs, Chen et al. (2019) propose L-Shapley and C-Shapley for efficient approximation of Shapley values, with a similar hypothesis with us that the importance of a word is usually only strongly dependent on its neighboring contexts. Chen et al. (2018) propose a feature selection based approach for explanation in an information theoretic perspective.

On the other hand, global explanation algorithms (Guidotti et al., 2018) have also been studied for identifying generally important features, such as Feature Importance Ranking Measure (Zien et al., 2009), Accumulated Local Effects (Apley, 2016), while usually restricted in the domain of tabular data. We note that the context independence property in our proposed methods implies we study hierarchical explanation as a global explanation problem (Guidotti et al., 2018). Compared with local explanation algorithms, global explanation algorithms are less studied for explaining individual predictions (Poerner et al., 2018), because they reveal the average behavior of models. However, with a hierarchical organization, we show global explanations are also powerful at explaining individual predictions, achieving better human evaluation scores and could explain compositional semantics where local explanation algorithms such as additive feature attribution algorithms totally fail. Moreover, we note that the use of explanation algorithms is not exclusive; we may apply explanation algorithms of different categories to make a more holistic explanation of model predictions.

Another closely related field is statistical feature interaction detection (Hooker, 2004; Sorokina et al., 2008; Tsang et al., 2017) from learned models, which usually focus on tabular data only. An exception is (Tsang et al., 2018), which also studies word interactions in neural sequence model predictions, but does not study interactions in a phrase level.

## 6 CONCLUSION

In this work, we identify two desirable properties for informative hierarchical explanations of predictions, namely the non-additivity and context-independence. We propose a formulation to quantify context independent importance of words and phrases that satisfies the properties above. We revisit the prior line of works on contextual decomposition algorithms, and propose Sampling and Contextual Decomposition (SCD) algorithm and Sampling and Occlusion algorithm (SOC). Experiments on multiple datasets and models show that our explanation algorithms generate informative hierarchical explanations, help to extract classification rules from models, and enhance human trust of models.

## ACKNOWLEDGEMENTS

This research is based upon work supported in part by NSF SMA 18-29268 and JP Morgan AI Research Award. We would like to thank all the collaborators in USC INK research lab for their constructive feedback on the work.

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

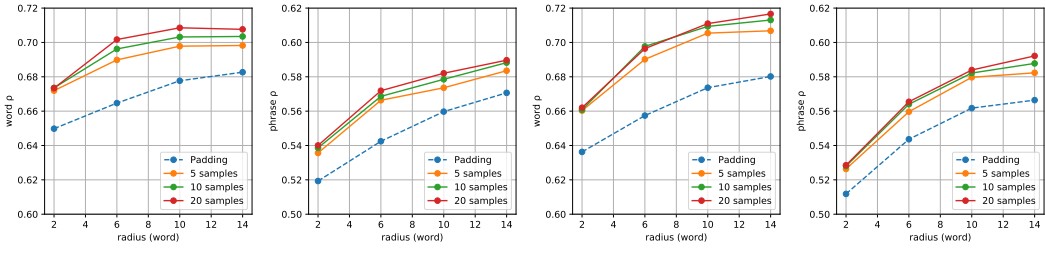

(a) SCD, word $\rho$, LSTM  (b) SCD, phrase $\rho$, LSTM  (c) SOC, word $\rho$, LSTM  (d) SOC, phrase $\rho$, LSTM

Figure 7: Word $\rho$ and phrase $\rho$ curve as the size of the context region $N$ and the number of samples $K$ change on LSTM trained on the SST-2 dataset. Dash line notes for the performance of padding the context words instead of sampling.

## A  IMPLEMENTATION DETAILS

Our LSTM classifiers use 1 layer unidirectional LSTM and the number of hidden units is set to 128, 500, and 300 for SST-2, Yelp, and TACRED dataset respectively. For all models, we load the pretrained 300-dimensional Glove word vectors (Pennington et al., 2014). The language model sampler is also built on LSTM and have the same parameter settings as the classifiers. Our Transformer models are fine-tuned from pretrained BERT models (Devlin et al., 2018), which have 12 layers and 768 hidden units of per representation. On three datasets, LSTM models achieve 82% accuracy, 95% accuracy, and 0.64 F1 score on average. The fine-tuned BERT models achieve 92% accuracy, 96% accuracy, and 0.68 F1 score on average. We use the same parameter settings between LSTM classifiers and language models on three datasets. Following Murdoch et al. (2018), we randomly sample a subset (set to 500 instances) of sentences of length at most 40 words for explanation on Yelp dataset, and also for TACRED dataset. On TACRED dataset, we generate explanations only for correctly predicted instances with a label other than no_relation.

## B  PERFORMANCE ON ADVERSARIAL MODELS

For computing context independent importance of a phrase, an intuitive and simple alternative approach, which is nevertheless neglected in prior literature, is to only feed the input to the model and treat the prediction score as the explanation. In Table 1, while the score of the Direct Feed is lower than that of the best performing algorithms, the score is rather competitive.

The potential risk of this explanation is that it assumes model performs reasonably on incomplete sentence fragments that are significantly out of the data distribution. As a result, the explanation of short phrases can be misleading. To simulate the situation, we train a LSTM model on inversed labels on isolate words, in addition to the original training instances. The model could achieve the same accuracy as the original LSTM model. However, the word $\rho$ and the phrase $\rho$ of Direct Feed drop by a large margin, showing a word $\rho$ of -0.38 and 0.09. SOC and SCD are still robust on the adverse LSTM model, both showing a word $\rho$ and phrase $\rho$ of more than 0.60 and 0.55.

The masking operation could also cause performance drop because the masked sentence can be out of data distribution when explaining long phrases. For SOC, the risk can be resolved by implementing the masking operation of the phrase $\mathbf{p}$ by another round of sampling from a language model conditioned on its context $\mathbf{x}_{-\mathbf{p}}$, but we do not find empirical evidence showing that it improves performance.

## C  EXPLANATION HEATMAPS

| Model | BERT | | LSTM | |
|---|---|---|---|---|
| Metric | word $\rho_{\mathrm{avg}}$ | phrase $\rho_{\mathrm{avg}}$ | word $\rho_{\mathrm{avg}}$ | phrase $\rho_{\mathrm{avg}}$ |
| Input Occlusion | 0.2947 | 0.4365 | 0.7150 | 0.4968 |
| GradSHAP | 0.5444 | 0.6078 | 0.7360 | 0.5449 |
| CD | 0.3642 | 0.3755 | 0.6962 | 0.4777 |
| ACD | 0.3023 | 0.3913 | 0.3210 | 0.2110 |
| SCD | 0.5601 | 0.6072 | 0.7348 | **0.5685** |
| SOC | **0.6282** | **0.6706** | **0.7430** | 0.5675 |

Table 2: Correlation between **averaged** word & phrase importance attribution and linear model coefficients and human annotations over all 2210 test instances in SST-2 dataset. The relative performance is similar to the case without score averaging as shown in Table 1.

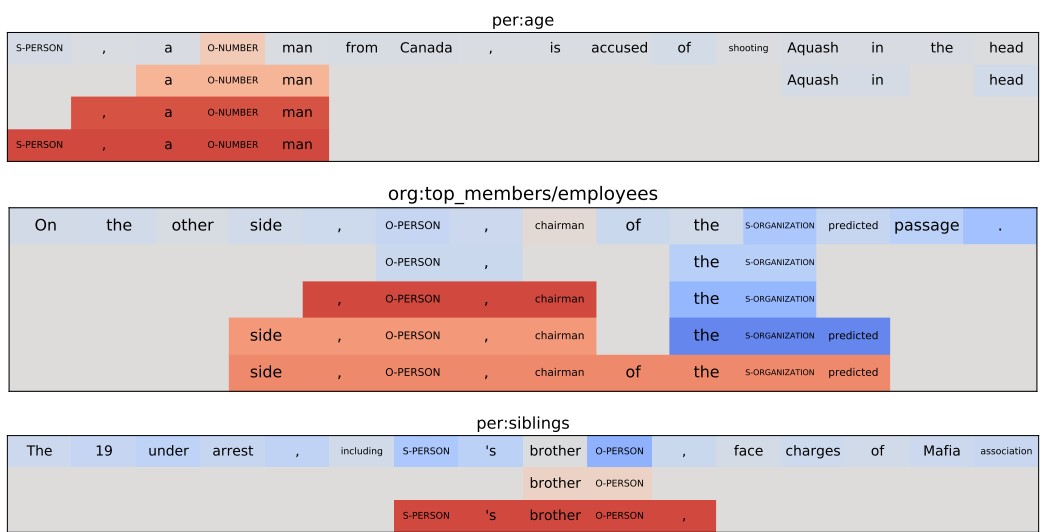

Figure 8: More examples about rule extraction from LSTM models trained on TACRED relation extraction dataset with SOC. Red indicate evidence for predicting the class, and blue indicate distractor for predicting the class. By applying the agglomerative clustering algorithm and defining a threshold score, we effectively extract classification rules from LSTM models. The ground truth label noted on the top

| Dataset | Label | Pattern |
|---|---|---|
| SST-2 | Positive | frighteningly evocative; insight and honesty |
| | Negative | neither funny nor provocative; kill the suspense |
| TACRED | person:age | [PERSON], [NUMBER], was; a [NUMBER] man |
| | organization:top-member | chief engineer of the [ORGANIZATION] |
| | person:origin | [Nationality] citizen; tribal member from [COUNTRY] |

Table 3: Phrase-level classification patterns extracted from models. We show the results of SCD and SOC respectively for the SST-2 and the TACRED dataset.

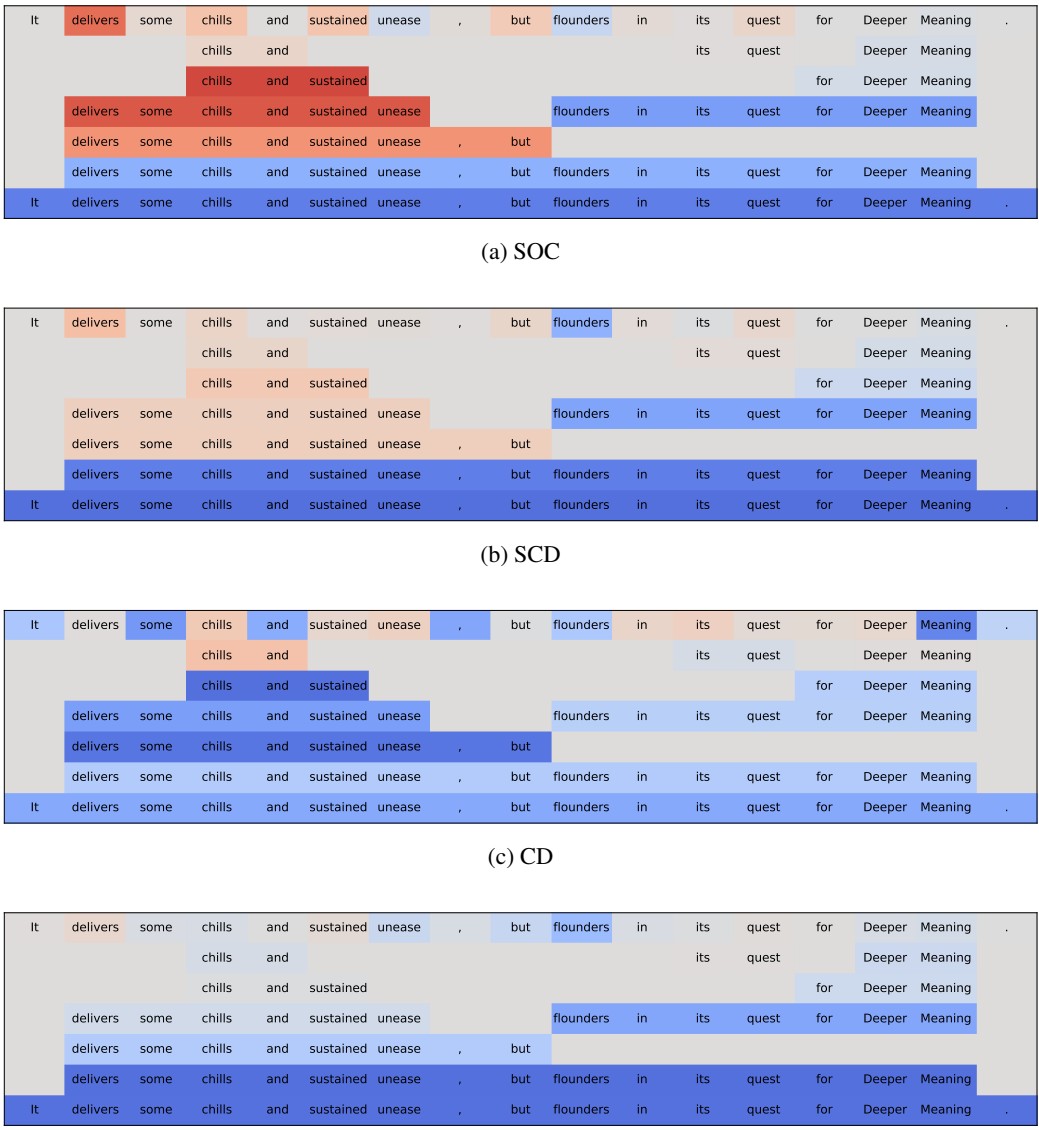

Figure 9: Explanation heatmaps generated by SOC, SCD, CD, and GradSHAP on a negatively predicted sentence by BERT Transformer model in SST-2 dataset. Only SOC and SCD captures adversarial conjunction connected by "but".

