# OpenReview forum: "Towards Hierarchical Importance Attribution: Explaining Compositional Semantics for Neural Sequence Models"
_ICLR.cc/2020/Conference — Accept (Spotlight)_

### Official Review · AnonReviewer1 · 2019-10-24
**Official Blind Review #1**

**Rating:** 8

**Review:**

Summary - The paper addresses the problem of hierarchical explanations in deep models that handle compositional semantics of words and phrases. The paper first highlights desirable properties for importance attribution scores in hierarchical explanations, specifically, non-additivity and context independence, and shows how prior work on additive feature attribution and context decomposition doesn’t accurately capture these notions. After highlighting the said properties in context of related work, the authors propose an approach to calculate the context-independent importance of a phrase by computing the difference in scores with and without masking out the phrase marginalized over all possible surrounding word contexts (approximated by sampling surrounding context for a fixed radius under a language model). Furthermore, based on the above, the authors propose two more score attribution approaches -- based on integrating the above sampling step with (1) the contextual decomposition pipeline and (2) the input occlusion pipeline. Experimentally, the authors find that the attribution scores assigned by the proposed approach are more correlated with human annotations compared to prior approaches and additionally, the generated explanations turn out to be more trustworthy when humans evaluate their quality.

Strengths

- The paper is well-written and generally easy to follow. The authors do a good job of motivating and highlighting the desired properties of importance attribution scores and developing the proposed scoring mechanism. The proposed scoring mechanism ties in seamlessly with the existing contextual decomposition and occlusion pipelines and leads to improved performance when the generated explanations are evaluated.

- The proposed approach involving masking out the phrase and marginalizing over possible surrounding word-concepts is novel and offers an interesting perspective on how to approach context independent scoring of phrases -- (1) phrases don’t exist independent of the surrounding context and therefore marginalizing over all possible surrounding concepts makes sense and (2) replacing the intractable enumeration over all possible surrounding concepts with samples from a language model makes the score attribution process faster and more scalable modulo the learnt language model.

- Sec. 4.4 offers interesting insights. I like that the authors performed this ablation given that the expectation over surrounding contexts is computed approximately via samples under a language model. There’s a clear increase in terms of the attribution scores as the number of samples increases and the neighborhood size is increased. It is interesting to note that there is an approaching plateau region where increasing the neighborhood size won’t affect the assigned scores. This experiment provides a holistic picture of the behavior of the interpretability toolkit (manifesting in terms of attribution scores) given the approximations involved. I would encourage the authors to flesh this out even more.

Weaknesses

Having said that, there are some minor comments that I’d like to point out / get the authors’ opinion on. Highlighting these below:

- While SOC and SCD don’t always end up outperforming other approaches (specifically Statistic) on the SST-2, Yelp and TACRED datasets (Table. 1), for the human evaluation experiments, the authors only compare with CD, Direct Feed, ACD and GradSHAP. Do the authors have any insights on how well does Statistic perform on the human-evaluation set of experiments?

- While inspiring trust in users is one aspect of evaluating explanations via humans, it’s slightly unclear what ‘trust’ in this sense inherently identifies. Although, it might implicitly capture some notion of reliability (and predictability of the explanations by humans), asking users to rank explanations across a spectrum of ‘best’ to ‘worst’ doesn’t explicitly capture that. Another possible aspect to look into could be -- ‘’Do the generated explanations help humans predict the output of the model?’’ This captures reliability in a very explicit sense. Do the authors have any thoughts on this and potential experiments that might address this? I don’t think not addressing this is necessarily detrimental to the paper but I’m curious to hear the thoughts of the authors on the same.

Reasons for rating

Beyond the above points of discussion, I don’t have major weaknesses to point 	out. I generally like the paper. The authors do a good job of identifying the sliver in which they make their contribution and motivate the same appropriately. The proposed phrase attribution scoring mechanism is motivated from a novel perspective and has a reasonable approximation characterized appropriately by the ablations performed. The strengths and weaknesses highlighted above form the basis of my rating.

**Experience Assessment:**

I have read many papers in this area.

**Review Assessment: Checking Correctness Of Derivations And Theory:**

I assessed the sensibility of the derivations and theory.

**Review Assessment: Checking Correctness Of Experiments:**

I assessed the sensibility of the experiments.

**Review Assessment: Thoroughness In Paper Reading:**

I read the paper thoroughly.

---

> ### Author Response · Authors · 2019-11-11
> **Response to Review #1**
>
> Thank you for your careful evaluation of the paper and encouraging comments. Here are our responses to the questions.
>
> -------------------------------
> Q1: SOC and SCD don’t always end up outperforming other approaches (specifically Statistic) on the SST-2, Yelp and TACRED datasets (Table. 1).
>
> A1: “Statistic” is a direct approximation of the context independent importance defined in Eq.3 by sampling over the dataset. Therefore, the good performance of Statistic is actually an evidence for the effectiveness of our proposed formulation. However, one drawback of the Statistic based approach is that it works only for words and phrases that appear frequently in the dataset, which is usually not the case especially for long phrases. It is verified by the evaluation results shown in Table 1, where Statistic performs competitively on word level explanations, but perform poorly on phrase level explanations, with only a small improvement over input occlusion. It motivates us to sample the contexts of a phrase from a trained language model.
>
> Q1 (cont.): For the human evaluation experiments, dothe authors have any insights on how well does Statistic perform on the human-evaluation set of experiments?
>
> Regarding human evaluation experiments, we limited the number of presented explanations to 4, mainly in consideration of reducing the difficulties for evaluators. Therefore, we did not include Statistic in human experiments. Nevertheless, according to the evaluation results and analysis based on Table 1, we hypothesis that Statistic based approach would perform inferiorly compared to SOC and SCD in phrase level explanations.
>
> -------------------------------
> Q2: Another possible aspect to look into human evaluation could be -- ‘’Do the generated explanations help humans predict the output of the model?’’ This captures reliability in a very explicit sense. Do the authors have any thoughts on this and potential experiments that might address this? I don’t think not addressing this is necessarily detrimental to the paper but I’m curious to hear the thoughts of the authors on the same.
>
> A2: Yes, we agree that we may explore other human evaluation protocols. We find the suggested human evaluation protocol quite consistent with a popular definition of interpretability: “the degree to which a human can consistently predict the model’s result”[1]. Thank you for the suggestion and we will consider this protocol in our future research.
>
> [1] Molnar, Christoph. "Interpretable machine learning", 2019.

---

> > ### Comment · AnonReviewer1 · 2019-11-15
> > **Thanks for responding to the comments.**
> >
> > Thanks to the authors for responding to my comments in detail.
> >
> > - Thanks for clarifying the concerns around the performance of "Statistic" approach.
> >
> > - Thanks for mentioning the reasons behind not using "Statistic" in the human-studies. My concern was primarily motivated by the performance of "Statistic" in the results. It seems reasonable to assume one would consider including "Statistic" in the human-studies as well. My only basis as of now is the hypothesis provided by the authors regarding this.

---

### Official Review · AnonReviewer4 · 2019-10-26
**Official Blind Review #4**

**Rating:** 6

**Review:**

Summary:
The authors proposed a method for generating hierarchical importance attribution for any neural sequence models (LSTM, BERT, etc.) Towards this goal, the authors propose two desired properties: 1) non-additivity, which means the importance of a phrase should be a non-linear function over the importance of its component words; 2) context independence, which means that the attribution of any given phrase should be independent of its context. For example, in the sentence "the film is not interesting", the attribution of "interesting" should be positive while the attribution of "not interesting" should be negative.

Following these two properties, the authors designed three algorithms to post-hoc analysis the importance of a given phrase p.
1. [Sec 3.2] eq 4. expected differences in model predictions between the a sentence that contains p and the same sentence with p removed. The expectation is computed over the conditional probability Prob(sentence | p in sentence). In practice, the authors use eq 3 as a proxy to eq 4.
2. [Sec 3.3] eq 5.  expected differences in the activation values of each layer. The expectation is computed over the conditional probability Prob(context-dependent representations | phrase-dependent representations).
3. [Sec 3.4] eq 8. similar to 1 but we replace the phrase p with padded tokens.

The authors conducted experiments on SST and Yelp. Results show that their proposed context-independent attribution correlates better with a trained linear model's coefficient, achieves higher human trust.

Decision: reject.

While I found the idea of marginalizing out the local context interesting, I think the paper still needs more work on its formulation, experiments and writing.

Formulation:
1. In eq 3, the expectation is taken over the difference between the prediction on the sampled sentence and the one with the phrase removed. This may be problematic for longer inputs (a pargraph), where the overall prediction may not change a lot when you remove a single phrase (since the evidence is everywhere). For example, consider the input: "The movie is the best that I have ever seen. It is remarkable!". Removing the word "best" alone doesn't alter the prediction much.
2. In eq 5, the expectation is computed over P(h | beta). It is NOT THE SAME as sampling words p(x_{\delta} | x_{-\delta}) and then consider their hidden states.
3. In Sec 3.1, you mentioned that CD is limited since the decomposition of activation sigma evolves context information gamma, and you resolved this by marginalization. But it seems to me that the computation of element wise multiplication also evolves context information. How do you deal with these?
3. What's the difference between eq3 and eq8? Are you just changing from remove the phrase completely to replace it by mask?

Experiments:
1. The performance of CD in Table 1 seems very different to the original CD paper (which is 0.758 for SST and 0.520 for Yelp). I am not sure what contributes to this big difference. Is it the trained model or data splits?
2. Table 1 shows that your methods achieves higher correlation to linear model's coefficients. But why shall we consider linear model's coefficients as the ground truth for the learned neural model? For example, the fine-tuned BERT achieves lower correlation corresponding to the LSTM. Does that mean the BERT model performs worse than LSTM?

Missing related references:
1. Explaining Image Classifiers by Counterfactual Generation
2. Rationalizing Neural Predictions
3. Learning to Explain: An Information-Theoretic Perspective on Model Interpretation
4. L-Shapley and C-Shapley: Efficient Model Interpretation for Structured Data

**Experience Assessment:**

I have published one or two papers in this area.

**Review Assessment: Checking Correctness Of Derivations And Theory:**

I carefully checked the derivations and theory.

**Review Assessment: Checking Correctness Of Experiments:**

I carefully checked the experiments.

**Review Assessment: Thoroughness In Paper Reading:**

I read the paper thoroughly.

---

> ### Author Response · Authors · 2019-11-11
> **Response to Review #4 (2/2)**
>
> Q5: What contributes to the differences between the reported results in CD paper and ours?
>
> Thank you for your careful look into the experimental results. For the CD algorithm, we use the official code released at [1] to ensure the correctness of the implementation. We also use standard data splits for training, validation, and testing, and evaluate our explanation algorithms on test set predictions. Therefore, we believe the difference is caused by the difference between the models used for reporting their results and ours. Nevertheless, we find that our algorithms perform competitively on different models: for example, by using the released code of CD for training the models on SST-2 dataset, we get a word correlation score of ~0.567 for CD, and ~0.697 for SOC.
>
> -------------------------------
> Q6: Regarding metrics selected in Table 1
>
> Quantitative evaluations of explanations are believed to be challenging as there are hardly any ground truths for explanations; the evaluations have to rely on hypothesis on what a neural network may have captured. We believe a good practice to evaluate neural network explanations is to compare it with multiple reference word or phrase importance. We chose coefficients of linear models as one of these references, because these coefficients are believed to be representative of word importance when the linear model is sufficiently accurate. We select the evaluation protocol also in consideration that it is reported in the CD paper.
>
> It is notable that in addition to the correlation with the linear model coefficients, we also tested phrase importance correlation with human annotations, as well as performed human evaluations and qualitative studies. Having all these results showing great consistency, we believe the effectiveness of our algorithms is justified by our experiments.
>
> -------------------------------
> Missing related references:
>
> Thank you for your pointers on related works. We have included them with some discussion in the updated related works section. Please check the updated version.
>
>  [1] https://github.com/jamie-murdoch/ContextualDecomposition

---

> > ### Comment · AnonReviewer4 · 2019-11-15
> > **Thanks for responding**
> >
> > The authors comments have addressed my concerns. I will raise the score.

---

> ### Author Response · Authors · 2019-11-11
> **Response to Review #4 (1/2)**
>
> We very much appreciate your careful and valuable comments!  Please find our detailed response is as follows.
>
> -------------------------------
> Q1: In Eq. 3 (context independent importance), the expectation is taken over the difference between the prediction on the sampled sentence and the one with the phrase removed. Will it lack of sensitivity when there are multiple evidences saturating the prediction? For example, consider the input: "The movie is the best that I have ever seen. It is remarkable!". Removing the word "best" alone doesn't alter the prediction much.
>
> A1: It is a great question. We agree that importance measured by removing a phrase lacks sensitivity when there are multiple evidences saturating the prediction *given one specific input*. However, the problem is greatly alleviated in Eq.3 by sampling input sequences $x$ from $p(x| p \subseteq x)$, as these drawn samples may have non-saturating predictions. For the given example in the question, our algorithms may evaluate the importance of the word “best” at some sampled input sequences where the word “best” is the only evidence, such as “the movie is the best that I have ever seen”. In this way, the proposed context-independent importance is robust to saturation.
>
> We also note that our formulation is robust to saturation in a similar way to other explanation algorithms, such as Shapley values. These algorithms average word importance given different subsets of context words in a specific input, to cover the case when the prediction is not saturated; our formulation is more general as it evaluates word/phrase importance given all possible contexts, weighted by their probability at the input space.
>
> Then, for the N-context independent importance, we utilized the assumption that a phrase usually strongly interacts with its neighboring contexts. The parameter analysis on N in section 4.4 shows the assumption generally holds true. Meanwhile, we also acknowledged at the end of section 3.4 that it is possible to extend SOC by applying other measures of phrase importance that are less affected by saturation in place of the input occlusion. We think it would potentially be helpful for longer input sequences. We have expanded the discussion at the end of section 3.4 in the new version of the paper.
>
> -------------------------------
> Q2: In eq 5, the expectation is computed over P(h | beta). It is NOT THE SAME as sampling words p(x_{\delta} | x_{-\delta}) and then consider their hidden states.
>
> A2: We appreciate the reviewer's careful look into the formulation. However, we did not mean $h$ is drawn from $p(h | \beta)$, and we believe the text part has caused this confusion. In the updated paper, at the beginning of Section 3.3 (after Eq. 5), we explicitly state $h$ is calculated from the sampled input sequences conditioned on $x_{-\delta}$. To clarify, in CD, $\beta^\prime$ terms are calculated as average activation differences after subtracting $\beta$, for the activation values $h$ computed on the specific input $x$ and that when $h=\beta$; SCD follows general protocols of CD and also calculates $\beta^\prime$ terms as average activation difference after subtracting $\beta$ terms, but for the activation values $h$ computed on each sampled input sequence.
>
> -------------------------------
> Q3: How does SCD deal with element-wise multiplication?
>
> A3: We treat element-wise multiplication the same as other activation functions, as stated at the end of section 3.3. More specifically, for $h = h_1 * h_2 $, the $\beta^\prime$ term is computed as $E_{\gamma_1, \gamma_2}[(\beta_1 + \gamma_1) * (\beta_2 + \gamma_2) - \gamma_1 * \gamma_2]$. We have added the formulation at the end of section 3.3 to improve the clarity.
>
> -------------------------------
> Q4: What's the difference between Eq. 3 and Eq. 8?
>
> A4: To clarify, Eq.3 introduces a general form of context independent importance, where the masking operation $x\backslash p$ should be defined in specific explanation algorithms. Eq.8 shows our SOC algorithm, where the masking operation implemented as replacing the given phrase with padding tokens.

---

### Official Review · AnonReviewer2 · 2019-10-27
**Official Blind Review #2**

**Rating:** 6

**Review:**

This paper proposes a hierarchical decomposition method to encode the natural language as mathematical formulation such that the properties of the words and phrases can encoded properly and their importance be preserved independent of the context. This formulation is intuitive and more efficient compared to blindly learning contextual information in the model. The proposed method is a modification of contextual decomposition algorithm by adding a sampling step. They also adapt the proposed sampling method into input occlusion algorithm as another variant of their method. The proposed method is tested on LSTM and BERT models over sentiment datasets of Stanford Sentiment Treebank-2 and Yelp Sentiment Polarity and TACRED relation extraction dataset and showed more interpretable generated hierarchical explanations compared to baselines.


**Experience Assessment:**

I have published one or two papers in this area.

**Review Assessment: Checking Correctness Of Derivations And Theory:**

I assessed the sensibility of the derivations and theory.

**Review Assessment: Checking Correctness Of Experiments:**

I assessed the sensibility of the experiments.

**Review Assessment: Thoroughness In Paper Reading:**

I read the paper at least twice and used my best judgement in assessing the paper.

---

> ### Author Response · Authors · 2019-11-11
> **Response to Review #2**
>
> Thank you very much for your comments! We will carefully revise and improve the draft for the final version. We believe the major contributions of our paper is to propose a formulation for measuring context independent importance, and proposed two explanation algorithms derived from the formulation. Please also refer to our responses to the other two reviewers if you have any further questions.

---

### Author Response · Authors · 2019-11-11
**Paper revised**

We would like to thank all the reviewers for their efforts and valuable comments. We have revised the paper to address the questions from reviewers, and also added additional ablation study. The major updates are:

- Added more related works in section 5, as suggested by reviewer #4.
- Added additional ablation study in section 4.4. We show padding the context gets inferior performance compared to sampling the context.
- Improved clarity in section 3.3-3.4, in response to reviewer #4.

---

### Decision · Program_Chairs · 2019-12-19

**Decision:**

Accept (Spotlight)

**Comment:**

The authors present a hierarchical explanation model for understanding the underlying representations produced by LSTMs and Transformers.  Using human evaluation, they find that their explanations are better, which could lead to better trust of these opaque models.

The reviewers raised some issues with the derivations, but the author response addressed most of these.